# Orthogonal Gromov-Wasserstein Discrepancy with Efficient Lower Bound

**Hongwei Jin**[1]                **Zishun Yu**[1]                **Xinhua Zhang**[1]

[1]Computer Science Department, University of Illinois at Chicago, Chicago, Illinois, USA

## Abstract

Comparing structured data from possibly different metric-measure spaces is a fundamental task in machine learning, with applications in, e.g., graph classification. The Gromov-Wasserstein (GW) discrepancy formulates a coupling between the structured data based on optimal transportation, tackling the incomparability between different structures by aligning the intra-relational geometries. Although efficient *local* solvers such as conditional gradient and Sinkhorn are available, the inherent non-convexity still prevents a tractable evaluation, and the existing lower bounds are not tight enough for practical use. To address this issue, we take inspirations from the connection with the quadratic assignment problem, and propose the orthogonal Gromov-Wasserstein (OGW) discrepancy as a surrogate of GW. It admits an efficient and *closed-form* lower bound with $\mathcal{O}(n^3)$ complexity, and directly extends to the fused Gromov-Wasserstein (FGW) distance, incorporating node features into the coupling. Extensive experiments on both the synthetic and real-world datasets show the tightness of our lower bounds, and both OGW and its lower bounds efficiently deliver accurate predictions and satisfactory barycenters for graph sets.

## 1 INTRODUCTION

Similarity based learning has been a popular approach in many machine learning applications. Instead of directly modeling each individual object which may pose challenge in some areas, it resorts to models based on pairwise similarity, possibly across different domains. The most common example is the kernels used in support vector machines and Gaussian process, including RBF kernel and covariance matrices that measure similarity in the Euclidean space [Chang et al., 2010], and string or tree kernels that compare discrete objects [Lodhi et al., 2002]. More generally, featured graphs have been a useful tool for capturing similarities and relations in the structured data that are commonly not Euclidean. Examples include social network [Fan et al., 2019], recommendation systems [Wu et al., 2020], fraud detection [Li et al., 2020], quantum chemistry [Coley et al., 2019] and topology-aware IoT applications [Abusnaina et al., 2019].

Despite the inherent challenge in comparing graphs from possibly different metric-measure spaces, there has been a wealth of refined discrepancy measures between graphs, including kernels [Vishwanathan et al., 2010, Shervashidze et al., 2011] and GCNs based approaches [Bronstein et al., 2017, Defferrard et al., 2016]. Recently, the Gromov-Wasserstein discrepancy [GW, Peyre et al., 2016], which extends the Gromov-Wasserstein distance [Memoli, 2011], has emerged as an effective transportation distance between structured data, alleviating the incomparability issue between different structures by aligning the *intra*-relational geometries. Thanks to its favorable properties such as efficiency and isometry-awareness, GW has been applied to domain adaptation [Yan et al., 2018], word embedding [Alvarez-Melis and Jaakkola, 2018], graph classification [Vayer et al., 2019a], metric alignment [Ezuz et al., 2017], generative modeling [Cohen and Sejdinovic, 2019], and graph matching and node embedding [Xu et al., 2019b,a].

However, different from the standard Wasserstein distance which is a linear program, GW is unfortunately intractable to evaluate. Despite the practical success of non-convex optimization techniques such as conditional gradient method and entropic regularization [Peyre et al., 2016, Gold and Rangarajan, 1996], it remains NP-hard to find the global optimum. Hence, existing practice settles with local solutions, lacking an analyzable guarantee. This significantly challenges the trustworthiness of the GW discrepancy.

Towards a tractable approximation, Memoli [2011] proposed three lower bounds of GW, which cost $O(n^2)$, $O(n^4)$ and $O(n^5)$. They can be useful for branch-and-bound based

*Accepted for the 39th Conference on Uncertainty in Artificial Intelligence* (UAI 2022).

global optimization, as well as recent algorithms for certifying the robustness of nonconvex models. However, these lower bounds are rarely used in practice, and as we will show later in Figure 3, even the most expensive lower bound can be quite loose, raising the concerns on their effectiveness.

Instead of developing yet another lower bound or tight approximation of GW, our goal in this paper is to design a *surrogate* of it (namely orthogonal GW, or OGW) such that:

(a) It retains GW's desirable properties such as permutation invariance, non-negativity, triangle inequality, and good performance in machine learning tasks such as classification and barycenter. We stress that OGW does not need to be either an upper bound or a lower bound of GW. We are also *not* concerned about the gap between OGW and GW because what matters is the desirable mathematical properties and the performance in learning, instead of how close OGW is to GW.

(b) It does *not* have to be tractable, but it must admit a *tight and efficient* lower bound – upper bound is easier from local optimization as in GW. The tightness will potentially contribute to global optimization such as certification, a topic that is beyond this paper. Ideally, such a lower bound should also possess the aforementioned good properties of the surrogate itself.

Our inspiration, as unrolled in Section 2, stems from the connection of GW to quadratic assignment problems (QAPs), which was tapped into by sliced GW [Vayer et al., 2019b] and Gromov-Monge problems [Memoli and Needham, 2021]. It paved the way for approximating the set of doubly stochastic matrices (used in GW) by orthonormal matrices under marginal constraints [Rendl and Wolkowicz, 1992, Hadley et al., 1992, Anstreicher and Brixius, 2001]. The resulting problem is well known to admit tight approximate solutions, and accommodates fused GW to account for node features [FGW, Vayer et al., 2019a]. Experiments on classification and barycenter demonstrate the effectiveness of our proposed OGW and its lower bounds.

## 2 OGW-DISCREPANCY WITH TRACTABLE LOWER BOUNDS

We represent an undirected graph $G$ with $n$ nodes by an adjacency matrix $\mathbf{A} \in \{0, 1\}^{n \times n}$, where $\mathbf{A}_{ii} = 0$, and $\mathbf{A}_{ij} = 1$ if there is an edge between nodes $i$ and $j$ ($i \neq j$), and 0 otherwise. To start with, we consider the discrepancy between two graphs with the same order (*i.e.*, number of nodes). A detailed generalization to any graph orders will be addressed in Section 2.3. Associated with the nodes is a distribution, encoding some prior information about their importance, e.g., the normalized degree of each node [Xu et al., 2019b]. However, many applications lack such natural normalization [Vayer et al., 2019a,b, Peyre et al., 2016].

Therefore, we will stick with a uniform distribution over all nodes and include our extension of non-uniform distribution in Appendix A.5. Letting $\mathbf{1} = (1, \ldots, 1)^\top$ whose dimensionality can be implicitly induced from the context, the standard GW distance between graph $G$ and $H$ based on $\ell_2$ distance can be formulated as (the square root of)

$$\mathsf{GW}(G, H) :=$$
$$\min_{\mathbf{P} \in \mathcal{E} \cap \mathcal{N}} \sum_{i,j,k,l} [c_G(i,j) - c_H(k,l)]^2 \, \mathbf{P}_{ik} \mathbf{P}_{jl}, \quad (1)$$

where $\mathcal{E} := \{\mathbf{P} \in \mathbb{R}^{n \times n} : \mathbf{P1} = \mathbf{P}^\top \mathbf{1} = \mathbf{1}\}$ and $\mathcal{N} := \mathbb{R}_+^{n \times n}$ [Memoli, 2011]. Here $c_G(i,j)$ represents a distance measure between node $i$ and $j$ on $G$, and common choices include their shortest-path distance, or simply $1 - \mathbf{A}_{ij}$ (the complement of adjacency). When both $c_G$ and $c_H$ are a metric, (1) is a squared metric on isomorphism classes of measurable metric spaces. However, as pointed out by Peyre et al. [2016], $c_G$ and $c_H$ do not have to be restricted to metrics and $\ell_2$ can be extended to other asymmetric or non-subadditive losses such as $f$-divergence. They call it GW discrepancy, which broadens its applicability in machine learning. We will refer to it as GW, without even taking the square root of (1) just like in Peyre et al. [2016].

**Remark 1.** *It is noteworthy that although the original GW requires $c_G(i,j)$ to be a distance metric [Memoli, 2011], it can be relaxed in (1) where $\ell_2$ loss is used. Indeed, $c_G(i,j)$ can also be served by* similarities *between nodes instead of distance, e.g., by simply flipping the sign of the distance measure. This also opens up the use of non-metric dissimilarity measures such as constrained shortest path [Lozano and Medaglia, 2013].*

Define two $n$-by-$n$ symmetric matrices $\mathbf{C}$ and $\mathbf{D}$ whose $(i,j)$-th elements are $\mathbf{C}_{ij} = c_G(i,j)$ and $\mathbf{D}_{ij} = c_H(i,j)$, respectively. For example, the complement of adjacency can be written as $\mathbf{C} = \mathbf{11}^\top - \mathbf{A}$. The above GW can be compactly rewritten in the Koopmans-Beckmann form [Koopmans and Beckmann, 1957]:

$$\mathsf{GW}(G, H) =$$
$$\frac{1}{n^2} \left( \|\mathbf{C}\|_F^2 + \|\mathbf{D}\|_F^2 - 2 \max_{\mathbf{P} \in \mathcal{E} \cap \mathcal{N}} \mathrm{tr}(\mathbf{CPDP}^\top) \right). \quad (2)$$

Here $\|\cdot\|_F$ is the Frobenius norm. Obviously, GW is permutation invariant, nonnegative, and equals 0 when $G$ and $H$ are isomorphic. The major drawback is that the maximization over $\mathbf{P}$ is intractable, although efficient *local* algorithms are available such as conditional gradient [Vayer et al., 2019a] and Sinkhorn [Peyre et al., 2016].

### 2.1 CONNECTING OGW WITH THE QUADRATIC ASSIGNMENT PROBLEM

Rewriting GW with quadratic optimization over $\mathcal{E} \cap \mathcal{N}$ as in (2) reveals an innate connection to the quadratic assignment

$$\begin{array}{ccc} \text{GW} & \xleftarrow{\hspace{1cm}} \quad \text{QAP} \quad \xrightarrow{\hspace{1cm}} & \text{OGW} \\ \mathcal{E} \cap \mathcal{N} & \mathcal{E} \cap \mathcal{N} \cap \mathcal{O} & \mathcal{E} \cap \mathcal{O} \end{array}$$

Figure 1: Connection between QAP and GW, OGW

problem (QAP). Noting that by the Birkhoff–von Neumann theorem [Birkhoff, 1946], $\mathcal{E} \cap \mathcal{N}$ is the convex hull of the set of $n \times n$ permutation matrices (denoted as $\Pi$). Indeed, the connection with QAP has been used to formulate Gromov-Monge distances [Memoli and Needham, 2021], and to accelerate the evaluation of GW via projection (slicing) to 1-D [Vayer et al., 2019b]. Fortunately, a number of tractable relaxations of QAP are available, many of which are based on the following characterization of $\Pi$:

$$\Pi = \mathcal{E} \cap \mathcal{N} \cap \mathcal{O}, \tag{3}$$

where $\mathcal{O} := \{\mathbf{P} \in \mathbb{R}^{n \times n} : \mathbf{P}^\top \mathbf{P} = \mathbf{P}\mathbf{P}^\top = \mathbf{I}\}$. Here $\mathbf{I}$ is the identity matrix. Whenever necessary, we will explicitize the dimensionality of $\mathcal{O}$ by writing $\mathcal{O}_n$. Interestingly, $\mathrm{tr}(\mathbf{CPDP}^\top)$ can be maximized exactly by a simple eigen-decomposition if $\mathbf{P}$ is restricted to $\mathcal{O}$ [Umeyama, 1988]. Specifically, assume the eigen-decomposition of $\mathbf{C}$ and $\mathbf{D}$ are $\mathbf{C} = \mathbf{P_C} \mathrm{diag}(\boldsymbol{\lambda_C})\mathbf{P_C}^\top$ and $\mathbf{D} = \mathbf{P_D} \mathrm{diag}(\boldsymbol{\lambda_D})\mathbf{P_D}^\top$, respectively, and suppose the eigenvalues in $\boldsymbol{\lambda_C}$ and $\boldsymbol{\lambda_D}$ are both arranged in a descending order. Then

$$\mathbf{P_1}\mathbf{P_2}^\top \in \arg\max_{\mathbf{P} \in \mathcal{O}} \mathrm{tr}(\mathbf{CPDP}^\top), \tag{4}$$

$$\text{and} \quad \max_{\mathbf{P} \in \mathcal{O}} \mathrm{tr}(\mathbf{CPDP}^\top) = \boldsymbol{\lambda_C}^\top \boldsymbol{\lambda_D}. \tag{5}$$

Based on this result, Hadley et al. [1992] proposed tightening the domain approximation from $\mathcal{O}$ to $\mathcal{O} \cap \mathcal{E}$, which, despite the original intention of approximating inhomogeneous QAPs, happens to be useful in our context too. Compared with $\mathcal{N} \cap \mathcal{E}$, $\mathcal{O} \cap \mathcal{E}$ offers more convenience in constructing upper and lower bounds that are tight and efficient. This substitution leads to our proposed new metric, named as *orthogonal Gromov-Wasserstein (OGW)* discrepancy:

$$\text{OGW}(G, H) :=$$
$$\frac{1}{n^2}\left(\|\mathbf{C}\|_F^2 + \|\mathbf{D}\|_F^2 - 2\max_{\mathbf{P} \in \mathcal{O} \cap \mathcal{E}} \mathrm{tr}(\mathbf{CPDP}^\top)\right). \tag{6}$$

Figure 1 illustrates how QAP is connected with GW and OGW through the different convex outer approximations of the domain of the permutation matrices.

## 2.2 UPPER AND LOWER BOUNDS OF OGW

The evaluation of OGW is hindered by the nonconvex objective *and* the nonconvex domain in the optimization of $\mathbf{P}$ in (6). So it is natural to resort to its lower and upper bounds.

**Upper bound of OGW.** Obviously, any locally optimal $\mathbf{P}$ in (6) yields an upper bound of OGW. To ease the local

optimization, we first leverage the characterization of $\mathcal{O} \cap \mathcal{E}$ [Hadley et al., 1992]:

$$\mathcal{O} \cap \mathcal{E} = \left\{\tfrac{1}{n}\mathbf{1}\mathbf{1}^\top + \mathbf{V}\mathbf{Q}\mathbf{V}^\top : \mathbf{Q} \in \mathcal{O}_{n-1}\right\}, \tag{7}$$

where $\mathbf{V}$ is any $n \times (n-1)$ matrix satisfying $\mathbf{V}^\top \mathbf{1} = \mathbf{0}$ and $\mathbf{V}^\top \mathbf{V} = \mathbf{I}_{n-1}$. An example is given in Appendix A.1. Plugging $\mathbf{P} = \frac{1}{n}\mathbf{1}\mathbf{1}^\top + \mathbf{V}\mathbf{Q}\mathbf{V}^\top$ into the optimization objective in (6) yields

$$\max_{\mathbf{P} \in \mathcal{O} \cap \mathcal{E}} \mathrm{tr}(\mathbf{CPDP}^\top) =$$
$$\frac{1}{n^2}s_\mathbf{C}s_\mathbf{D} + \underbrace{\max_{\mathbf{Q} \in \mathcal{O}} \{\mathrm{tr}(\hat{\mathbf{C}}\mathbf{Q}\hat{\mathbf{D}}\mathbf{Q}^\top) + \mathrm{tr}(\hat{\mathbf{E}}^\top \mathbf{Q})\}}_{=: \, \mathcal{Q}(\hat{\mathbf{C}}, \hat{\mathbf{D}}, \hat{\mathbf{E}})}, \tag{8}$$

where $\hat{\mathbf{X}} := \mathbf{V}^\top \mathbf{X}\mathbf{V}$ and $s_\mathbf{X} := \mathbf{1}^\top \mathbf{X}\mathbf{1}$ for any matrix $\mathbf{X}$, and $\mathbf{E} := \frac{2}{n}\mathbf{C}\mathbf{1}\mathbf{1}^\top \mathbf{D}$. Since $\mathcal{Q}(\hat{\mathbf{C}}, \hat{\mathbf{D}}, \hat{\mathbf{E}})$ involves both linear and quadratic terms in $\mathbf{Q}$, no closed-form solution remains available.

Clearly, any locally optimal $\mathbf{Q}$ yields a *lower bound* for $\mathcal{Q}$ (denoted as $\mathcal{Q}_{lb}$), *i.e.*, an upper bound for OGW (denoted as $\text{OGW}_{ub}$). *Locally* optimizing $\mathbf{Q}$ over $\mathcal{O}$ (*a.k.a.* Stiefel manifold) has been very well studied [Absil et al., 2009, Wen and Yin, 2013, Arasu and Mohan, 2018], and we adopt a straightforward approach of projected quasi-Newton, noting that the projection of any matrix $\mathbf{Q}$ on $\mathcal{O}$ is simply $\mathbf{U_Q}\mathbf{V_Q}^\top$, where the singular value decomposition (SVD) of $\mathbf{Q}$ is $\mathbf{U_Q}\Lambda_\mathbf{Q}\mathbf{V_Q}^\top$. With the locally optimal $\mathbf{Q}$ in hand, the locally optimal $\mathbf{P}$ for OGW can be recovered by plugging $\mathbf{Q}$ into the formula in (7).

Similarly to the practice of GW which resorts to locally optimal solutions, we will use $\text{OGW}_{ub}$ as a practical "evaluation" of OGW. Whenever there is no confusion (especially in empirical investigation), we will simply refer to the performance of $\text{OGW}_{ub}$ as the performance of OGW.

**Lower bounds of OGW.** The simplest way to lower bound OGW is by relaxing the domain of $\mathbf{P}$ into $\mathcal{O}$ in (6):

$$\text{OGW}_o(G, H)$$
$$:= \frac{1}{n^2}\left(\|\mathbf{C}\|_F^2 + \|\mathbf{D}\|_F^2 - 2\max_{\mathbf{P} \in \mathcal{O}} \mathrm{tr}(\mathbf{CPDP}^\top)\right) \tag{9}$$
$$= \frac{1}{n^2}\|\boldsymbol{\lambda_C} - \boldsymbol{\lambda_D}\|^2, \tag{10}$$

where the last step is by (4). We note in passing that $\text{OGW}_o$ embodies a different design principle from heat kernel signature [Sun et al., 2009] and wave kernel signature [Aubry et al., 2011], in that neither of the kernel signatures sort the kernel spectrum.

In practice, we found that completely dropping the constraint $\mathcal{E}$ may lead to over relaxation. To bring back $\mathcal{E}$, we follow Hadley et al. [1992] and decompose $\mathcal{Q}$ in (8) into

quadratic and linear terms by decoupling their $\mathbf{Q}$:

$$\mathcal{Q}_{ub}(\hat{\mathbf{C}}, \hat{\mathbf{D}}, \hat{\mathbf{E}})$$
$$:= \max_{\mathbf{Q}_1 \in \mathcal{O}} \mathrm{tr}(\hat{\mathbf{C}}\mathbf{Q}_1\hat{\mathbf{D}}\mathbf{Q}_1^\top) + \max_{\mathbf{Q}_2 \in \mathcal{O}} \mathrm{tr}(\hat{\mathbf{E}}^\top\mathbf{Q}_2). \quad (11)$$

As a result, we obtain an *upper bound* of $\mathcal{Q}$ (denoted as $\mathcal{Q}_{ub}$), which produces a *lower bound* of OGW:

$$\mathsf{OGW}_{lb}(G, H) := \quad (12)$$
$$\frac{1}{n^2}\left(\|\mathbf{C}\|_F^2 + \|\mathbf{D}\|_F^2 - 2\,\mathcal{Q}_{ub}(\hat{\mathbf{C}}, \hat{\mathbf{D}}, \hat{\mathbf{E}}) - \frac{1}{n^2}s_{\mathbf{C}}s_{\mathbf{D}}\right).$$

$\mathcal{Q}_{ub}$ can be evaluated analytically. First, $\mathbf{Q}_1$ can be solved by (4). As for $\mathbf{Q}_2$, the von Neumann's trace inequality implies that its optimal value is $\mathbf{U_E}\mathbf{V_E}^\top$, where $\mathbf{U_E}\mathbf{\Lambda_E}\mathbf{V_E}^\top$ is the SVD of $\hat{\mathbf{E}}$, and the maximum value of $\mathrm{tr}(\hat{\mathbf{E}}^\top\mathbf{Q}_2)$ is $\|\hat{\mathbf{E}}\|_*$, the trace norm of $\hat{\mathbf{E}}$, which is the sum of the singular values of $\hat{\mathbf{E}}$. Hadley et al. [1992] showed that such an upper bound in (11) is often quite tight, which is also observed in our experiments. Indeed, we noticed that the magnitude of $\hat{\mathbf{C}}$ and $\hat{\mathbf{D}}$ in (11) is significantly larger than that of $\hat{\mathbf{E}}$. Therefore, although the optimal $\mathbf{Q}_1$ and $\mathbf{Q}_2$ are different, the resulting gap is small.

We next summarize the mathematical properties of OGW and its lower bounds as follows:

**Theorem 1.** *OGW, $\mathsf{OGW}_o$, and $\mathsf{OGW}_{lb}$ are all nonnegative and symmetric. Their square root satisfies the triangle inequality. Their values are 0 if (but not only if) the two graphs are isomorphic.*

The proof is in Appendix A.2. Compared with the requirement of distance metric, OGW and its lower bounds only fall short of the "only if" part of the identity of indiscernibles. To see why "only if" cannot hold, consider $\mathsf{OGW}_o$ whose closed form in (9) shows that its value can be 0 as long as $\mathbf{C}$ and $\mathbf{D}$ are similar, *i.e.*, share the same eigenvalues. In general, however, $\mathbf{C}$ and $\mathbf{D}$ are derived from graphs with certain *discrete* properties, leaving permutation the most likely path to similarity.

**Remark 2.** *The coupling matrix $\mathbf{P}$ in GW provides a useful matching between two sets of nodes. Although the $\mathbf{P}$ in OGW and its lower bounds may contain negative entries, it optimizes over the orthonormal domain, which may still provide useful insights between the two groups of node. For example, invariance to orthogonal transformation is a longstanding pursuit in learning [Kornblith et al., 2019]. Despite the hardness of exactly optimizing $\mathbf{P}$ for OGW, we can use $\mathbf{Q}_1$ from (11) to recover $\mathbf{P}$ via the transformation in (7). This is reasonable because $\hat{\mathbf{E}}$ is generally much smaller in magnitude than $\hat{\mathbf{C}}$ and $\hat{\mathbf{D}}$.*

**Computational complexity.** The analytic solution for $\mathsf{OGW}_{lb}$ and $\mathsf{OGW}_o$ is achieved by singular value decomposition and eigen decomposition, whose computational complexity is $\mathcal{O}(n^3)$. $\hat{\mathbf{C}}$, $\hat{\mathbf{D}}$, and $\hat{\mathbf{E}}$ can be computed in $\mathcal{O}(n^3)$ thanks to the structure of $\mathbf{V}$ (see Appendix A.1). It is worth mentioning that Memoli [2011] also derived the lower bounds of GW by solving a set of linear assignment problems, named First Lower Bound (FLB), Second Lower Bound (SLB), and Third Lower Bound (TLB). The following inequalities provide the connections between different lower bounds of GW:

$$\mathsf{GW} \geq \begin{cases} \mathsf{GW}_{tlb} \geq \mathsf{GW}_{flb} \\ \mathsf{GW}_{slb}. \end{cases} \quad (13)$$

And the complexities of FLB, SLB and TLB are $\mathcal{O}(n^2), \mathcal{O}(n^4), \mathcal{O}(n^5)$, respectively. In general, TLB provides the tightest lower bound for GW, and SLB has resemblant performance compared with TLB. This is also observed in our experiments.

## 2.3 GRAPHS WITH DIFFERENT SIZES

So far, we have been restricting the two graphs to have the same number of nodes, primarily because the orthonormal domain $\mathcal{O}$ only contains square matrices. In order to deal with the non-square matrix, *i.e.*, graphs of different order, we introduce the semi-orthogonal domain

$$\tilde{\mathcal{O}}_{m,n} := \{\mathbf{T} \in \mathbb{R}^{m \times n} : \mathbf{T}^\top\mathbf{T} = \mathbf{I}_n\}, \quad (14)$$

where $m > n$ without loss of generality. That is a domain of "tall" matrices, whose columns are orthonormal. When $m = n$, $\tilde{\mathcal{O}}$ recovers $\mathcal{O}_n$ because $\mathbf{T}^\top\mathbf{T} = \mathbf{I}_n$ is equivalent to $\mathbf{T}\mathbf{T}^\top = \mathbf{I}_n$ for a square matrix $\mathbf{T}$.

For a non-square matrix $\mathbf{P} \in \mathbb{R}^{m \times n}$ and $m > n$, it is no longer feasible to impose the constraint of $\mathcal{E} := \{\mathbf{P} \in \mathbb{R}^{n \times n} : \mathbf{P1} = \mathbf{P}^\top\mathbf{1} = \mathbf{1}\}$ because the dimensionality does not match. Instead, it can be generalized into

$$\tilde{\mathcal{E}} := \{\mathbf{P1}_n = \sqrt{\tfrac{n}{m}}\mathbf{1}_m, \mathbf{P}^\top\mathbf{1}_m = \sqrt{\tfrac{m}{n}}\mathbf{1}_n\}. \quad (15)$$

To summarize, we can extend the OGW in (6) to graphs with different orders by replacing the domain of $\mathbf{P}$, amounting to

$$\mathsf{OGW}(G, H) := \quad (16)$$
$$\frac{1}{m^2}\|\mathbf{C}\|_F^2 + \frac{1}{n^2}\|\mathbf{D}\|_F^2 - \frac{2}{mn}\max_{\mathbf{P} \in \tilde{\mathcal{O}} \cap \tilde{\mathcal{E}}}\mathrm{tr}(\mathbf{CPDP}^\top),$$

where $G$ and $H$ have the graph order of $m$ and $n$, respectively. We reuse the symbol OGW because the constraints $\tilde{\mathcal{O}}$ and $\tilde{\mathcal{E}}$ recover $\mathcal{O}$ and $\mathcal{E}$ respectively when $m = n$. It is also easy to see that OGW is nonnegative because

$$\mathsf{OGW}(G, H)$$
$$\geq \frac{1}{m^2}\|\mathbf{C}\|_F^2 + \frac{1}{n^2}\|\mathbf{D}\|_F^2 - \frac{2}{mn}\max_{\mathbf{P} \in \tilde{\mathcal{O}}}\mathrm{tr}(\mathbf{CPDP}^\top)$$
$$= \left\|\tfrac{1}{m}\boldsymbol{\lambda}_{\mathbf{C}} - \tfrac{1}{n}\boldsymbol{\lambda}_{\mathbf{D}}\right\|^2 \geq 0.$$

In the similar spirit to (7), any $\mathbf{P} \in \tilde{\mathcal{O}} \cap \tilde{\mathcal{E}}$ can be reparameterized as follows

**Theorem 2.** *Let $\mathbf{U} \in \mathbb{R}^{m \times (m-1)}$ and $\mathbf{V} \in \mathbb{R}^{n \times (n-1)}$ be arbitrary projection matrices satisfying*

$$\mathbf{U}^\top \mathbf{1}_m = \mathbf{0}_{m-1}, \quad \mathbf{V}^\top \mathbf{1}_n = \mathbf{0}_{n-1}, \qquad (17)$$

$$\mathbf{U}^\top \mathbf{U} = \mathbf{I}_{m-1}, \quad \mathbf{V}^\top \mathbf{V} = \mathbf{I}_{n-1}. \qquad (18)$$

*Then*

$$\mathbf{P} \in \tilde{\mathcal{O}} \cap \tilde{\mathcal{E}} \iff \mathbf{P} = \frac{1}{\sqrt{mn}} \mathbf{1}_m \mathbf{1}_n^\top + \mathbf{U} \mathbf{Q} \mathbf{V}^\top, \quad (19)$$

*where $\mathbf{Q} \in \tilde{\mathcal{O}}_{m-1, n-1}$.*

The proof is relegated to Appendix A.4. Letting $\hat{\mathbf{C}} := \mathbf{U}^\top \mathbf{C} \mathbf{U}$ and $\hat{\mathbf{D}} := \mathbf{V}^\top \mathbf{D} \mathbf{V}$, the optimization over $\mathbf{P}$ in (16) turns into a projected QAP (PQAP) with an additional linear term and some constant terms

$$\max_{\mathbf{Q} \in \tilde{\mathcal{O}}_{m-1, n-1}} \text{tr}(\hat{\mathbf{C}} \mathbf{Q} \hat{\mathbf{D}} \mathbf{Q}^\top) + \text{tr}(\hat{\mathbf{E}} \mathbf{Q}^\top) + \text{const}, \quad (20)$$

where $\hat{\mathbf{E}} = \frac{2}{\sqrt{mn}} \mathbf{U}^\top \mathbf{C} \mathbf{1}_m \mathbf{1}_n^\top \mathbf{D} \mathbf{V}$. Finally, in order to leverage the favorable properties of orthonormal matrices, we right-pad the matrix $\mathbf{Q}$ by an $(m-1) \times (n-m)$ matrix $\mathbf{J}$, such that $[\mathbf{Q}, \mathbf{J}] \in \mathcal{O}_{m-1}$. Indeed

$$\mathbf{Q} \in \tilde{\mathcal{O}}_{(m-1),(n-1)} \iff \exists \mathbf{J} : \begin{bmatrix} \mathbf{Q} & \mathbf{J} \end{bmatrix} \in \mathcal{O}_{m-1}. \quad (21)$$

Such a $\mathbf{J}$ matrix only needs to be any basis of the kernel space of $\mathbf{Q}$. As a result, the quadratic term $\max_{\mathbf{Q} \in \tilde{\mathcal{O}}} \text{tr}(\hat{\mathbf{C}} \mathbf{Q} \hat{\mathbf{D}} \mathbf{Q}^\top)$ is equivalent to

$$\max_{[\mathbf{Q} \ \mathbf{J}] \in \mathcal{O}} \text{tr}\left( \hat{\mathbf{C}} \begin{bmatrix} \mathbf{Q} & \mathbf{J} \end{bmatrix} \begin{bmatrix} \hat{\mathbf{D}} & \mathbf{0} \\ \mathbf{0} & \mathbf{0} \end{bmatrix} \begin{bmatrix} \mathbf{Q} & \mathbf{J} \end{bmatrix}^\top \right), \quad (22)$$

and we can optimize $[\mathbf{Q} \ \mathbf{J}]$ as a whole, utilizing the closed-form solution to the squared matrix case.

Similarly, for the linear term, we have

$$\max_{\mathbf{Q} \in \tilde{\mathcal{O}}} \text{tr}(\hat{\mathbf{E}} \mathbf{Q}^\top) = \max_{[\mathbf{Q} \ \mathbf{K}] \in \mathcal{O}} \text{tr}\left( \begin{bmatrix} \hat{\mathbf{E}} & \mathbf{0} \end{bmatrix} \begin{bmatrix} \mathbf{Q} & \mathbf{K} \end{bmatrix}^\top \right), (23)$$

where $\mathbf{K}$ serves the same role as $\mathbf{J}$.

To conclude, by padding zero on the non-square matrices $\hat{\mathbf{D}}$ and $\hat{\mathbf{E}}$ to square matrices, we can enjoy the analytic solutions to the problems in (22) and (23) in the same way as in the square case.

**Remark 3.** *We refrain from the interpretation of adding dummy nodes with 0 distance because, as pointed out in Remark 1, $\mathbf{C}$ and $\mathbf{D}$ can represent similarity measures. In such cases, padding with 0 is still justified with the above derivation, but not amenable to dummy node interpretations.*

**Remark 4.** *Disconnected graphs can be modeled by any existing heuristic that is also required by GW. In (1), $c_G(i, j)$ cannot be $\infty$ because it would push GW to $\infty$ as long as all $c_H(k, l) < \infty$ and all nodes have nonzero marginals. A*

*simple heuristic is to employ a large distance value between two nodes that belong to two separate/disconnected subgraphs. Our experiments only involved connected graphs, because all the graphs from the real datasets are already connected – none was discarded.*

We stress that the heuristic in Remark 3 is independent of padding 0 in (22) that tackles different graph sizes. The latter is on how to characterize the alignment of two matrices, which is orthogonal to the design of base measure itself within each graph.

## 3   BARYCENTER

We next study the application of OGW to the Barycenter problem, where given a set of sampled graphs and their associated weights, we aim to find their Fréchet mean by minimizing the weighted average discrepancy between the barycenter and the sampled graphs. Here the discrepancy is measured by our proposed OGW, and the sampled graphs are represented by a set of cost matrices $\mathbf{D}_i$, along with normalized weight $\lambda_i$ for $i \in \{1, ..., S\}$. $\mathbf{D}_i$ can also encode pairwise similarities instead of dissimilarities, and our method accommodates both cases naturally. The barycenter problem can be formalized as minimizing the weighted average OGW discrepancy:

$$\min_{\mathbf{C}} \mathcal{B}(\mathbf{C}) := \min \sum_{i=1}^{S} \lambda_i \cdot \text{OGW}(\mathbf{C}, \mathbf{D}_i) \qquad (24)$$

$$= \sum_{i=1}^{S} \lambda_i \left( \frac{1}{m^2} \|\mathbf{C}\|_F^2 - \frac{2}{m n_i} \max_{\mathbf{P}_i \in \mathcal{O} \cap \mathcal{E}} \text{tr}(\mathbf{C} \mathbf{P}_i \mathbf{D}_i \mathbf{P}_i^\top) \right)$$

$$+ \text{const}. \qquad (25)$$

For simplicity, we specify the barycenter with a fixed order $m$, although the value of $m$ can also be optimized.

We follow the block coordinate update proposed in Peyre et al. [2016], *i.e.*, iteratively minimizing with respect to the couplings $\mathbf{P}_i$ and updating the optimal cost matrix for the barycenter in a closed-form solution. Given a set of coupling matrices $\mathbf{P}_i$, we can retrieve the optimal $\mathbf{C}$ by taking the partial derivative of (24)

$$\frac{\partial \mathcal{B}(\mathbf{C})}{\partial \mathbf{C}} = \sum_i \lambda_i \left( \frac{2}{m^2} \mathbf{C} - \frac{2}{m n_i} \mathbf{P}_i \mathbf{D}_i \mathbf{P}_i^\top \right) \qquad (26)$$

$$= \frac{2}{m^2} \mathbf{C} - \sum_i \frac{2\lambda_i}{m n_i} \mathbf{P}_i \mathbf{D}_i \mathbf{P}_i^\top = \mathbf{0}. \qquad (27)$$

So the optimal $\mathbf{C}$ under the current $\{\mathbf{P}_i\}$ is $\mathbf{C}^* \leftarrow m \sum_{i=1}^{S} \frac{\lambda_i}{n_i} \mathbf{P}_i \mathbf{D}_i \mathbf{P}_i^\top$. Noting that OGW itself is still intractable, we resort to the tractable lower or upper bounds, replacing OGW in the definition of $\mathcal{B}(\mathbf{C})$ with its tractable bounds. In the case of $\text{OGW}_{ub}$ (resp. $\text{OGW}_o$), we simply adopt the locally (resp. globally) optimal $\{\mathbf{P}_i\}$. For $\text{OGW}_{lb}$,

Figure 2: Steps of constructing the graph discrepancies

we can rewrite $\mathcal{B}(\mathbf{C})$ in terms of $\mathbf{Q}_1$ and $\mathbf{Q}_2$ from (11) and (12), and then $\mathbf{C}$ can be updated using the optimal $\mathbf{Q}_1$ and $\mathbf{Q}_2$. More details are provided in Appendix A.7.

It is noteworthy that any optimal solution $\mathbf{C}$ for (24) leads to a set of optimal solutions $\{\mathbf{PCP}^\top : \mathbf{P} \in \mathcal{O} \cap \mathcal{E}\}$. This also resonates with the intuition retrievable from $\mathsf{OGW}_o$ in (9), where only the eigenvalues of $\mathbf{C}$ matter. Therefore, additional "post-processing" is needed to pinpoint the optimal $\mathbf{C}$ from the equivalent class. Furthermore, the optimal $\mathbf{C}^*$ found over $\mathcal{O} \cap \mathcal{E}$ does not guarantee elementwise nonnegativity. Next, we present our method, named as *spectral reconstruction*, to find the appropriate $\mathbf{C}$.

### 3.1 SPECTRAL RECONSTRUCTION

To begin with, suppose $\mathbf{C}$ and all $\mathbf{D}_i$ are graphs of order $m$, and we consider their projections to $\mathbb{R}^{(m-1)\times(m-1)}$ via $\hat{\mathbf{C}}^* = \mathbf{V}^\top \mathbf{C}^* \mathbf{V}$ and $\hat{\mathbf{D}}_i = \mathbf{V}^\top \mathbf{D}_i \mathbf{V}$. Let their eigendecomposition be $\hat{\mathbf{C}}^* = R^\top \Sigma R$ and $\hat{\mathbf{D}}_i = S_i^\top \Delta_i S_i$. Inspired by the above observation that $\mathsf{OGW}(\mathbf{C}, \mathbf{D}_i)$ depends primarily on the eigenvalues of $\mathbf{C}$ and $\mathbf{D}_i$, we rebuild $\hat{\mathbf{C}}^*$ by $\hat{\mathbf{C}}_{recon}^* = \sum_i \lambda_i S_i^\top \Sigma S_i$, *i.e.*, trusting and retaining the eigenvalues of the optimal solution $\hat{\mathbf{C}}^*$ while pairing them with the eigenvectors of the sampled graphs.

For graphs with different sizes, the trick in Section 2.3, *i.e.*, padding on smaller graph, helps us to assemble the $\hat{\mathbf{C}}^*$ from the top $m-1$ eigen system. Noting that $\hat{\mathbf{C}}^*$ is still on the projected domain $\mathbb{R}^{m-1\times m-1}$, *i.e.*, there exist $\mathbf{V}$ such that $\hat{\mathbf{C}}^* = \mathbf{V}^\top \mathbf{C}^* \mathbf{V}$.

Next, we bring $\hat{\mathbf{C}}_{recon}^*$ back to its original space $\mathbb{R}^{m\times m}$ via

$$\mathbf{C}_{recon}^* = \mathbf{V}\hat{\mathbf{C}}_{recon}^* \mathbf{V}^\top + \mathbf{Y}, \qquad (28)$$

where $\mathbf{Y}$ satisfies $\mathbf{V}^\top \mathbf{Y} \mathbf{V} = \mathbf{0}$, ensuring that $\mathbf{V}^\top \mathbf{C}_{recon}^* \mathbf{V}$ recovers $\hat{\mathbf{C}}_{recon}^*$. In addition, when $\mathsf{OGW}$ operates on dissimilarity matrices, we require $\mathrm{diag}(\mathbf{C}_{recon}^*) = \mathbf{0}$, i.e., the dissimilarity between a node and itself is 0. A straightforward choice of $\mathbf{Y}$ satisfying the two conditions is

$$\mathbf{Y} = -\frac{1}{2}(\mathbf{d}\mathbf{1}^\top + \mathbf{1}\mathbf{d}^\top), \qquad (29)$$

$$\text{where} \quad \mathbf{d} := \mathrm{diag}(\mathbf{V}\hat{\mathbf{C}}_{recon}^* \mathbf{V}^\top). \qquad (30)$$

To gain more intuition into the recipe, consider the barycenter problem with $\mathsf{OGW}_{lb}$ and only one sampled graph $\mathbf{D}_1$.

By Theorem 1, $\mathcal{B}(\mathbf{C})$ can be driven to 0 and it is attained when $\hat{\mathbf{C}}$ shares the same eigenvalues as $\hat{\mathbf{D}}_1$, *i.e.*, $\Sigma = \Delta_1$. Then $\hat{\mathbf{C}}_{recon}^* = \hat{\mathbf{D}}_1$ by our construction. Furthermore, the proof of Theorem 1 indicates $\mathbf{1}^\top \mathbf{C}\mathbf{1} = \mathbf{1}^\top \mathbf{D}_1 \mathbf{1}$ and $\|V^\top \mathbf{C}\mathbf{1}\| = \|V^\top \mathbf{D}_1 \mathbf{1}\|$. It is then not hard to show that $\mathbf{C}_{recon}^*$ is exactly $\mathbf{D}_1$. We provide our experiments on both synthetic and real dataset in Section 5.3.

## 4 EXTENSION TO FUSED GW

Most applications carry features for each node. To account for this important information, Vayer et al. [2019a] proposed the fused GW (FGW), employing an additional matrix $\mathbf{M}$ whose $(i,k)$-th entry encodes the $\ell_2$ distance between the features of node $i$ in $G$ and of node $k$ in $H$. $\mathbf{M}$ is asymmetric in general. Then the vanilla FGW-distance was formulated by Vayer et al. [2019a] as

$$\mathsf{FGW}(G, H, \mathbf{M}) := \frac{\alpha}{n^2} \|\mathbf{C}\|_F^2 + \frac{\alpha}{n^2} \|\mathbf{D}\|_F^2 \qquad (31)$$
$$- \frac{1}{n^2} \max_{\mathbf{P} \in \mathcal{E} \cap \mathcal{N}} \left\{ 2\alpha \, \mathrm{tr}(\mathbf{CPDP}^\top) - (1-\alpha)\mathrm{tr}(\mathbf{M}^\top \mathbf{P}) \right\},$$

where $\alpha \in [0,1]$ is a trade-off between structure and feature measure. For simplicity, we will only present the treatment for two graphs of the same size. The extension to different sizes can be easily derived in the same way as in Section 2.3. Similar to $\mathcal{Q}$, with the additional linear term $\mathrm{tr}(\mathbf{M}^\top \mathbf{P})$, there is no closed-form solution even if we replace the domain of $\mathbf{P}$ by $\mathcal{O}$. However, we can still tighten the domain from $\mathcal{O}$ to $\mathcal{O} \cap \mathcal{E}$, leading to our new approximation

$$\mathsf{OFGW}(G, H, \mathbf{M}) := \frac{\alpha}{n^2} \|\mathbf{C}\|_F^2 + \frac{\alpha}{n^2} \|\mathbf{D}\|_F^2 \qquad (32)$$
$$- \frac{1}{n^2} \max_{\mathbf{P} \in \mathcal{E} \cap \mathcal{O}} \left\{ 2\alpha \, \mathrm{tr}(\mathbf{CPDP}^\top) - (1-\alpha)\mathrm{tr}(\mathbf{M}^\top \mathbf{P}) \right\}.$$

The pipeline of construction is illustrated in Figure 2. As $\alpha$ tends to zero, OFGW recovers OGW between only structures. A number of favorable properties are enjoyed by OFGW, which are summarized in Theorem 3 below (proof deferred to Appendix A.3). Although OGW must be nonnegative, OFGW is not guaranteed nonnegative for all $\mathbf{M}$. To see a counter-example, set $\mathbf{C} = \mathbf{D} = \mathbf{0}$ and $\mathbf{M} = \mathbf{I}$. Fortunately, for a large set of $\mathbf{M}$, it still enjoys nonnegativity.

**Theorem 3.** *Suppose $M$ satisfies $\mathbf{1}^\top \mathbf{M}\mathbf{1} \geq n\|\hat{\mathbf{M}}\|_*$. Then OFGW$(G, H, \mathbf{M}) \geq 0$ for all $G$, $H$, and is invariant to the (different) permutations of $G$ and $H$. When $\mathbf{M} = \mathbf{0}$, it degenerates to OGW.*

Since $\mathbf{M}$ encodes the pairwise distance between two sets of node features $\{\mathbf{X}_i\}_{i=1}^n$ and $\{\mathbf{Y}_i\}_{i=1}^n$ with $\mathbf{M}_{ij} = \|\mathbf{X}_i - \mathbf{Y}_j\|^2$, we can confirm whether the above assumption holds a priori. Interestingly, this is the case in all the datasets considered in our experiment. For datasets without node attributes and labels, we take node degree as their features. In the sequel, we will make this assumption on $\mathbf{M}$.

Although OFGW is still intractable in general, local optimization can be performed very efficiently. As will be shown in Section 4.1, it also admits a tight *lower bound* using the same relaxation technique as for OGW. Now that OFGW is motivated by computational convenience, one may naturally wonder whether it captures as much graph structure as the original FGW does. We verified this in the affirmative by following Vayer et al. [2019a], where FGW is used as a kernel function served in a support vector machine. In Section 5.1, we will show that replacing FGW by OFGW achieves similar or better classification accuracy on a variety of datasets, corroborating the effectiveness of OFGW.

## 4.1 UPPER AND LOWER BOUNDS OF OFGW

Following (7), plugging $\mathbf{P} = \frac{1}{n}\mathbf{1}\mathbf{1}^\top + \mathbf{V}\mathbf{Q}\mathbf{V}^\top$ into the optimization objective in (32) yields

$$\max_{\mathbf{P}\in\mathcal{O}\cap\mathcal{E}}\{2\alpha\operatorname{tr}(\mathbf{C}\mathbf{P}\mathbf{D}\mathbf{P}^\top) - (1-\alpha)\operatorname{tr}(\mathbf{M}^\top\mathbf{P})\} \qquad (33)$$
$$= \frac{1}{2n}s_{\mathbf{F}} - \frac{\alpha}{2n}s_{\mathbf{M}} + \underbrace{\max_{\mathbf{Q}\in\mathcal{O}}2\alpha\operatorname{tr}(\hat{\mathbf{C}}\mathbf{Q}\hat{\mathbf{D}}\mathbf{Q}^\top) + \operatorname{tr}(\hat{\mathbf{F}}^\top\mathbf{Q})}_{=:\,\mathcal{Q}(\hat{\mathbf{C}},\hat{\mathbf{D}},\hat{\mathbf{F}})},$$

where $\mathbf{F} := \frac{2\alpha}{n}\mathbf{C}\mathbf{1}\mathbf{1}^\top\mathbf{D} - \frac{1-\alpha}{2\alpha}\mathbf{M}$. It reveals that compared with the expression of GW in (8), the additional linear term $\mathbf{M}$ in the FGW formulation does not change the structure. As a result, we can again derive the *lower bound* of $\mathcal{Q}$, *i.e.*, *upper bound* of OFGW, by using the projected quasi-Newton as before. And a *lower bound* of OFGW can also be obtained by decoupling the $\mathbf{Q}$ in the two terms of (33). Specifically, by using the $\mathcal{Q}_{ub}$ defined (11) via decoupling into $\mathbf{Q}_1$ and $\mathbf{Q}_2$, we obtain

$$\mathsf{OFGW}_{lb}(G,H,\mathbf{M}) = \frac{\alpha}{n^2}\|\mathbf{C}\|_F^2 + \frac{\alpha}{n^2}\|\mathbf{D}\|_F^2$$
$$-\frac{1}{n^2}\left[2\alpha\,\mathcal{Q}_{ub}(\hat{\mathbf{C}},\hat{\mathbf{D}},\hat{\mathbf{F}}) + \frac{s_{\mathbf{F}}}{n} - \frac{1-\alpha}{2\alpha n}s_{\mathbf{M}}\right]. \qquad (34)$$

## 5 EXPERIMENTS

We now demonstrate the empirical effectiveness of OGW and OFGW via two applications: graph classification and barycenter problem. We will also illustrate the tightness of our lower bound OGW$_{lb}$. All the code and data are available at https://github.com/cshjin/ogw.

## 5.1 EFFECTIVENESS OF OGW/OFGW

Recall in Figure 2, $\mathcal{N}\cap\mathcal{E}$ was replaced by $\mathcal{O}\cap\mathcal{E}$ because the latter enjoys tight and efficiently computable upper and lower bounds. So it is important to validate the resulting OGW/OFGW as an equally good measure of comparing two graphs as the vanilla GW/FGW.

**Datasets.** We experimented on six graph classification datasets: BZR, COX2, MUTAG, PTC-MR, IMDB-Binary, and IMDB-Multi [TUDataset]. Their statistics are given in Appendix A.8. The first four datasets contain a collection of molecules (*e.g.*, chemical compound and ligands), where the vertices represent atoms and edges are chemical bonds. The class label represents a certain property of the molecules, *e.g.*, "mutagenic effect on a specific bacterium" (MUTAG) and carcinogenicity of compounds for male rats (PTC-MR). BZR and COX2 consist collections of ligands for the benzodiazepine receptor and cyclooxygenase-2 inhibitors, respectively. IMDB-Binary and IMDB-Multi are the movie collaboration dataset, where nodes represent actors/actresses who played roles in movies in IMDB, and one edge means two played in the same movie. We group the dataset into three categories according to their feature property: vectorized, discrete, and no features. For the datasets with no features, we take the node degrees as their features.

**Settings.** In order to evaluate a discrepancy measure $d$, we follow Vayer et al. [2019a] by studying the graph classification accuracy of an SVM, whose kernel $k(G,H)$ is computed by $\exp(-\gamma d(G,H))$. For both FGW and OFGW, the feature distance matrix $\mathbf{M}$ employed the squared Euclidean distance. Since OFGW itself is intractable to evaluate, we resort to the lower bound of OFGW in (31) which has an analytic form and is nonnegative. For the vanilla GW/FGW, we adopt the implementation from POT package [Flamary et al., 2021], which initiates the transition matrix by the outer product of marginal distributions. And we instantiate the cost matrix $c_G$ by all-pair shortest path for each graph in the datasets, knowing that the structures are all connected. We evaluate the models by cross-validation on the hyperparameters in SVM, setting $\gamma$ from $\{2^{-10}, 2^{-9}, \cdots, 2^{10}\}$ and $C$ from $2^{-4}$ to $2^4$ on evenly log scale with 15 steps. Moreover, for the FGW/OFGW, we cross-validate the value of $\alpha$ from $[0,1]$ with grid search.

In addition, we consider the graph kernel methods as the baselines. More specifically, we adopt the implementation of shortest path (SP) kernel [Borgwardt and Kriegel, 2005] and graphlet sampling (GK) kernel [Przulj, 2007] from Siglidis et al. [2020]. SP kernel decomposes graphs into shortest paths and compares pairs of shortest paths according to their lengths and the labels of their endpoint, while GK kernel decomposes graphs into graphlets, *i.e.*, small subgraphs with $k$ nodes where $k \in \{3,4,5,\cdots\}$, and counts matching graphlets in the input graphs.

**Results.** The average accuracy achieved with 10-fold cross-validation is presented in Table 1. To see the effectiveness of OGW and its lower bounds (OGW$_{lb}$ and OGW$_o$), we compare their accuracy against that of the vanilla GW and its first lower bound (GW$_{flb}$). In addition, we also include FGW and OFGW that incorporate node features. As the table shows, the tractable lower bounds of our proposed

Table 1: Graph classification

| | Dataset | Graph kernel | | GW-based SVM | | | OGW-based SVM | | | |
|---|---|---|---|---|---|---|---|---|---|---|
| | | SP | GK $(k=5)$ | GW | $GW_{flb}$ | FGW | $OGW_{ub}$ | $OGW_{lb}$ | $OGW_o$ | $OFGW_{lb}$ |
| Vec. | BZR | 78.8 ±3.3 | 78.8 ±3.3 | **84.9** ±**1.8** | 78.8 ±1.0 | 84.8 ±3.2 | 78.8 ±1.0 | 83.4 ±3.4 | 83.4 ±3.4 | 84.3 ±3.8 |
| Attr. | COX2 | 78.2 ±0.4 | 78.2 ±0.4 | 76.2 ±2.1 | 78.8 ±2.2 | 78.5 ±1.9 | 78.4 ±1.8 | 78.1 ±1.8 | 78.2 ±0.8 | **80.2** ±**2.4** |
| Disc. | MUTAG | 78.2 ±4.1 | 66.5 ±0.9 | 85.1 ±3.4 | 60.5 ±2.3 | **85.7** ±**2.4** | 66.5 ±2.3 | 82.8 ±3.0 | 82.8 ±3.0 | 85.4 ±1.7 |
| Attr. | PTC-MR | 57.3 ±1.0 | 55.8 ±0.7 | 53.4 ±4.3 | 60.2 ±5.1 | 51.8 ±3.4 | 59.5 ±10.1 | 57.9 ±4.5 | 57.9 ±4.5 | 57.1 ±4.1 |
| No | IMDB-B | 57.5 ±2.6 | 60.1 ±2.4 | 63.4 ±0.9 | 63.7 ±4.0 | 65.6 ±1.8 | 65.1 ±0.3 | **68.3** ±**1.7** | 67.4 ±1.1 | 67.3 ±2.1 |
| Attr. | IMDB-M | 39.7 ±1.8 | 38.2 ±2.7 | 47.5 ±2.3 | 43.2 ±2.6 | **49.7** ±**1.7** | 48.1 ±2.2 | 48.5 ±1.9 | 47.9 ±1.2 | 47.1 ±2.3 |

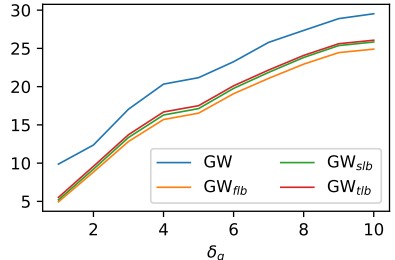
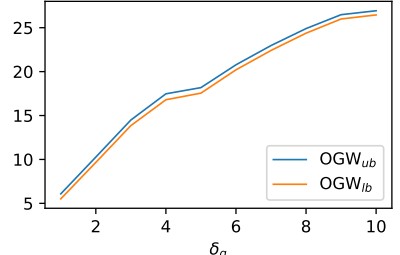
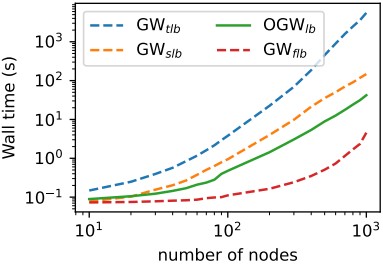

(a) Average value of GW and its lower bounds under $\delta_g$ number of perturbations

(b) Average value of $OGW_{ub}$ and $OGW_{lb}$ under $\delta_g$ number of perturbations

(c) Running time of the lower bounds

Figure 3: Average value of GW and OGW as compared with their respective lower bounds to demonstrate the tightness of the lower bounds. The running time of these lower bounds is also compared.

metric OGW have comparable performance with GW and its variant. With the additional information from node features (except IMDB datasets), the OFGW provides higher accuracy in general. It corroborates that the chain in Figure 2 preserves important structures in graphs, making the tractable lower bounds of OGW/OFGW sound discrepancy measures for graphs. Moreover, $OGW_{lb}$ achieves higher accuracy than $OGW_o$ on two datasets, and performs similarly on the other four datasets.

## 5.2 TIGHTNESS OF THE LOWER BOUND

The tightness of our lower bound $OGW_{lb}$, with respect to OGW, can be evaluated through its difference to the upper bound $OGW_{ub}$, which is obtained by the projected quasi-Newton method. Such a gap will be compared with its counterpart in GW distance, where the upper bound is served by a local optimizer based on the conditional gradient method, and the lower bounds are proposed by Memoli [2011], including FLB, SLB, and TLB (the best known lower bound).

**Synthetic data.** To demonstrate the tightness in synthetic data, we generate a path graph with 20 nodes and randomly perturb $\delta_g \in [1, 10]$ edges for 50 times, so that we can measure the distance (dissimilarity) between the original graph and the perturbed graph under different measures. Only connected graphs are kept. Figure 3a and 3b provide, respectively for GW and OGW, the average distance as a

function of the number of perturbed edges. The gap between $OGW_{ub}$ and $OGW_{lb}$ is much tighter than the best gap for the GW case, i.e., $GW - GW_{tlb}$.

Note that TLB of GW requires $\mathcal{O}(n^5)$ computational time, while $OGW_{lb}$ only costs $\mathcal{O}(n^3)$. To verify it, we measure the running time by varying the graph sizes. For each graph size that ranges from 10 to 1000, 20 Erdős-Rényi random graphs are generated, ensuring they are connected. Then their average running time is reported in Figure 3c, which clearly matches the analyzed computational complexity.

**Real-world data.** We also examine the gap between lower and upper bounds on the real-world dataset MUTAG by evaluating pairwise distance. Figure 4a provides the distribution of the gaps from GW and OGW. For GW, we report the gap between GW local optimizer and $GW_{tlb}$. Clearly, the gap between $OGW_{ub}$ and $OGW_{lb}$ centers around $0.1$, while that between the GW and its TLB concentrates around $1$.

In addition, we also compare in Figure 4b the tightness of $OGW_{lb}$ and $OGW_o$, both as a lower bound for OGW. Each point in the scatter plot represents a graph in the MUTAG dataset, and the horizontal (resp. vertical) axis is the gap between the upper bound of OGW and $OGW_{lb}$ (resp. $OGW_o$). The fact that the vast majority of the points lie above the diagonal confirms the superior tightness achieved by $OGW_{lb}$. This demonstrates the benefit of employing $\mathcal{E}$ in the constraint and separating $\mathbf{Q}_1$ and $\mathbf{Q}_2$ in (11). More tightness results on other datasets are provided in Appendix A.9.

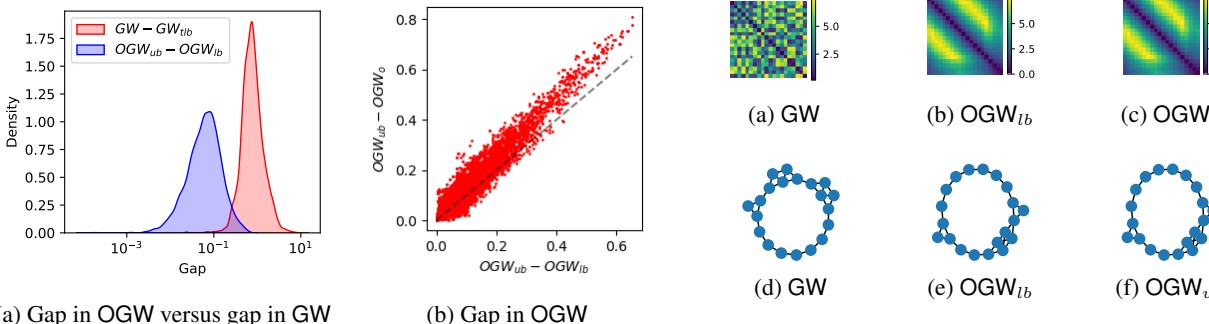

(a) Gap in OGW versus gap in GW  (b) Gap in OGW

Figure 4: Tightness of lower and upper bounds for GW and OGW (MUTAG dataset)

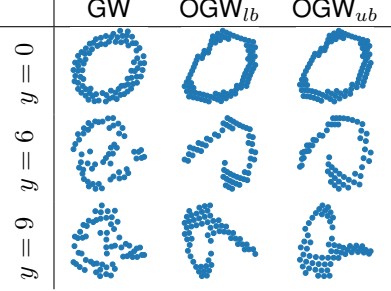

(a) GW  (b) OGW$_{lb}$  (c) OGW$_{ub}$

(d) GW  (e) OGW$_{lb}$  (f) OGW$_{ub}$

Figure 5: Optimized barycenter. Subfigures (a-c) show the $\mathbf{C}^*$ retrieved by optimizing the barycenter. Subfigures (d-f) show the reconstructed graph from the optimal cost matrix $\mathbf{C}^*$.

## 5.3 BARYCENTER

In the last set of experiments, we evaluate the ability of OGW$_{lb}$ and OGW$_{ub}$ to solve the barycenter problem.

**Synthetic data.** To start with, we generate a set of cycle-like graphs with different sizes ranging from 15 to 25. In addition, we also explicitly add random structural noise to the graphs separately, and pre-compute their shortest path as cost matrices. All synthetic samples are plotted in Appendix A.10. To initialize the barycenter, we fix the number of nodes to be 20 and initiate a random symmetric $\mathbf{C}$ when starting the block coordinate descend to update $\mathbf{P}$. For GW, we adopt the implementation from Peyre et al. [2016] to find the optimal cost matrix of the center. For the OGW$_{lb}$ and OGW$_{ub}$, we solve the problem by our proposed eigen projection method in Section 3.

To better visualize the results, we also reconstruct the adjacency matrix following a standard heuristic [Vayer et al., 2019a]. In particular, for a given threshold, a pair of nodes can be connected by an edge if and only if their corresponding entry in the cost matrix is below the threshold. Then we perform a line search on the threshold to minimize the difference between the optimal cost matrix and the one corresponding to the threshold based adjacency matrix. From Figure 5, we can see the reconstructed cost matrix for the barycenter from our OGW$_{lb}$ and OGW$_{ub}$ are well aligned to its node ordering. We also recognize that the result from GW is just one of the local minimizers in the context of permutation. Due to the tightness between the lower and upper bounds, it is also hard to differentiate the structures found by OGW$_{lb}$ and OGW$_{ub}$ (i.e., sub-figures e and f).

**Point cloud data.** We also explore the barycenter on point cloud dataset called MNIST-2D [1]. Refer to Appendix A.10 for the plots of the corresponding point cloud data. Clearly, the point clouds for 6 and 9 are quite similar up to rotation.

Table 2: Barycenters from point cloud MNIST-2D samples

| | GW | OGW$_{lb}$ | OGW$_{ub}$ |
|---|---|---|---|
| $y = 0$ | | | |
| $y = 6$ | | | |
| $y = 9$ | | | |

The point cloud is modeled by a graph whose nodes correspond to non-zero (non-black) pixels, represented by their 2D coordinates. Without constructing the explicit mesh connections between pixels, we take the Euclidean distance as their cost matrix $\mathbf{D}_i$. After retrieving the optimal $\mathbf{C}^*$ of the barycenter, we further uncover the associated optimal coordinates. A BFGS optimizer was used to seek the locally optimal coordinates such that the resulting Euclidean distance between pixels best reconstructs $\mathbf{C}^*$. Clearly, such a recovery can only be up to the standard invariant transformations such as rotation and shift, and they are determined by the random initialization of BFGS.

Table 2 illustrates the optimal point clouds with labels to be 0, 6, and 9. We sample 5 different point clouds for each digit and set the weight $\lambda_i$ to the uniform distribution. Moreover, we fix the number of nodes on the barycenter as the minimum size from the samples. Compared with GW, OGW$_{lb}$ and OGW$_{ub}$ clearly find better point clouds of the barycenter in the cases of digit 6 and 9. Moreover, thanks to the tight gap between the lower bounds, the performance of OGW$_{lb}$ and OGW$_{ub}$ differ only indistinguishably.

## Acknowledgements

We thank the reviewers for their constructive comments. This work is supported by NSF grant RI:1910146.

---

[1]Point cloud MNIST-2D dataset: https://www.kaggle.com/cristiangarcia/pointcloudmnist2d

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
