# OpenReview forum: "Orthogonal Gromov-Wasserstein Discrepancy with Efficient Lower Bound"
_auai.org/UAI/2022/Conference — UAI 2022 Poster_

### Official Review · Reviewer_hNew · 2022-03-31

**Q2(1) Originality/Novelty:** 2
**Q2(2) Significance/Impact:** 2
**Q2(3) Correctness/Technical Quality:** 3
**Q2(6) Clarity Of Writing:** 2
**Q6 Overall Score:** 5
**Q8 Confidence In Your Score:** 3

**Q1 Summary And Contributions:**

The authors propose the Orthogonal Gromov Wasserstein (OGW) discrepancy as an alternative to the GW distance, which results in non-convex objectives. OGW itself is also non-tractable so computable upper and lower bounds are given, with cost scaling like $O(N^3)$. Numerical examples show that the two bounds are closer, compared to tractable lower bounds for GW.

**Q2 Assessment Of The Paper:**

More detailed information regarding each of these aspects is given below:

**Q2(4) Quality Of Experiments (Optional):**

2: Fair: The experimental evaluation is weak: important baselines are missing, or the results do not adequately support the main claims.

**Q2(5) Reproducibility:**

3: Good: Key resources (e.g., proofs, code, data) are available and key details (e.g., proofs, experimental setup) are sufficiently well-described for competent researchers to confidently reproduce the main results.

**Q3 Main Strengths:**

The new distance does indeed admit upper and lower bounds which are quite close in the numerical experiments.

**Q4 Main Weakness:**

* The new discrepancy seems to be lacking motivation other than the fact that it makes calculations more tractable; does it's behaviour match our expectations of a distance between graphs let's say?
* There seem to be parts of the literature that have not been cited, e.g. arXiv:1906.02085 (neurips 2019), and arXiv:2106.01128
* The numerical experiments seem a little limited; it would be nice if the authors tried the method on more complicated datasets, e.g. the
PTC dataset or IMDB-B.

**Q5 Detailed Comments To The Authors:**

* Is OGW guaranteed to be non-negative for graphs with different sizes? This is not mentioned at all in section 2.3 and Theorem 1 refers specifically to graphs of the same order. Is it still 0 when graphs are isomorphic? It seems plausible this shouldn't be affected by the padding, but if that's the case why is it not stated more clearly? If it's not the case then this is an important limitation that should be clearly stated.
* For OFGW in the paragraph after equation 31 it is mentioned that OFGW is not guaranteed to be nonnegative, while OGW is guaranteed. Is this true also for graphs of different degrees? If so please consider restating THeorem 1 to include the general case.
* I find Section 2.4.1 extremely hard to follow.
       --The language needs improvement: e.g. "remedies" does not sound right; in Remark 1, "share the same set"
       --do the words semi-original domain, self-similarities mean something specific? has this been introduced?
* In Figure 3 please keep the y-axis scale the same between the first two plots to make comparisons easier.
* In eq. (8) use brackets to indicate the max applies to both Trace terms
* In appendix A, the norms are not constistently labelled with $_F$; for example in equation (40) we have $\|V^T C U\|_F$ where as in equation (38) it appears as $\|V^T C U\|$.
*Also in appendix A, it seems to me that the inequality leading to (39)-(41) is Young's inequality rather than the Cauchy-Schwartz inequality.

**Q7 Justification For Your Score:**

I am not convinced that the method is as useful or competitive as presented. More detailed comparisons with existing methods would be useful. Evidence that the proposed distance measures something reasonable should also be provided. Parts of the paper are not well writen.

**Q9 Complying With Reviewing Instructions:**

1: Yes.

---

### Official Review · Reviewer_m3YM · 2022-04-12

**Q2(1) Originality/Novelty:** 2
**Q2(2) Significance/Impact:** 2
**Q2(3) Correctness/Technical Quality:** 3
**Q2(6) Clarity Of Writing:** 3
**Q6 Overall Score:** 5
**Q8 Confidence In Your Score:** 3

**Q1 Summary And Contributions:**

The main motivation of this paper is the intractability of the GW distance. More
precisely, the authors extend the existing work on the relaxation of the Quadratic
Assignment Problem and propose the OGW distance, which admits a tractable, yet
quite tight lower bound.

**Q2 Assessment Of The Paper:**

More detailed information regarding each of these aspects is given below:

**Q2(4) Quality Of Experiments (Optional):**

3: Good: The experimental evaluation is adequate, and the results convincingly support the main claims.

**Q2(5) Reproducibility:**

3: Good: Key resources (e.g., proofs, code, data) are available and key details (e.g., proofs, experimental setup) are sufficiently well-described for competent researchers to confidently reproduce the main results.

**Q3 Main Strengths:**

The major novelty of this work:
- Providing preliminary theoritical analysis of OGW and FOGW.
- Extending both to the setting of different-size matrix.
- Proposing the eigen projection method to ensure the nonnegativity of the
barycenter matrix.

**Q4 Main Weakness:**

- advantages unclear in terms of practical interest
- the coupling matrix is no longer available

**Q5 Detailed Comments To The Authors:**

As the formulation and characterisation of OGW (namely equation 7,8) have already
been established from the work of (Harley et al.), I would expect, either more theorical
analysis of OGW, or more experiments on the performance of OGW vs other
methods.

While it is interesting to see the tightness between OGW/GW and its bounds (section
4.2), I would consider that such comparison is not very relevant and only the
tightness from OGW is necessary in the main paper. More precisely, in practice, OGW
is approximated from its bounds, so the tightness is important and useful. On the
other hand, the bounds of GW distance are rarely used (except that its FLB is
sometimes used as initialisation in the optimisation of GW distance). A much more
practically relevant lower bound is CO-Optimal transport (COOT), which is an even
tighter bound than TLB. Moreover, in some popular cases (euclidean or square
euclidean distance matrix), COOT = GW distance (so the bound is provably tight).
In the experiment 4.1, the authors claim that "Furthermore, it is noticeable that the
accuracies with GW in the dataset of COX2 and PTC-MR are slightly worse than those
in baselines. This is largely due to the intractable non-convex optimization with suboptimal
local minima in GW. Therefore, the tractable lower bound of OGW/OFGW are
sound discrepancy measures for graph threat models."
I am not convinced by this argument and I think it is speculative and overstated.
Based on only a few specific datasets, it is hard to deduce why one method can work
better than the others. In practice, the GW distance shows good performance, so the
intractablity or sub-optimality of the solution may not be that bad.
Another (minor) disadvantage of OGW distance is that the unique coupling matrix is
no longer available, which can limit its usage in practice.



Minor comments:

- Can the authors also illustrate the tightness of FOGW with its bounds?
- Can the authors elaborate more on the remark 1?
- Typo: in the theorem 2, \hat{M}  → M

**Q7 Justification For Your Score:**

I appreciate the effort of the authors on making OGW really usable in practice,
namely handling different size graphs, barycenter problem. On the other hand, the
tractability is attractive, but the practical evidence suggests that the intractablity of
GW distance may not be that bad.

**Q9 Complying With Reviewing Instructions:**

1: Yes.

---

### Official Review · Reviewer_6zCY · 2022-04-12

**Q2(1) Originality/Novelty:** 3
**Q2(2) Significance/Impact:** 3
**Q2(3) Correctness/Technical Quality:** 3
**Q2(6) Clarity Of Writing:** 3
**Q6 Overall Score:** 7
**Q8 Confidence In Your Score:** 4

**Q1 Summary And Contributions:**

The paper proposes an efficient lower bound for the orthogonal Gromov Wasserstein distance and shows its efficiency on several datasets.

**Q2 Assessment Of The Paper:**

More detailed information regarding each of these aspects is given below:

**Q2(4) Quality Of Experiments (Optional):**

3: Good: The experimental evaluation is adequate, and the results convincingly support the main claims.

**Q2(5) Reproducibility:**

3: Good: Key resources (e.g., proofs, code, data) are available and key details (e.g., proofs, experimental setup) are sufficiently well-described for competent researchers to confidently reproduce the main results.

**Q3 Main Strengths:**

The paper is well-written.
The paper provides a non-trivial and quite technically sound algorithmic and theoretical derivation
The paper has a set of illustrative experiments

**Q4 Main Weakness:**

No baselines. As a possible baseline (although it is an upper bound for a different problem) is Intrinsic Multiscale Distance (https://openreview.net/forum?id=HyebplHYwB) which could be compared to the upper bounds. This would strengthen the paper.
Computational complexity $O(n^3) $ seems to be quite high if n is large.

**Q5 Detailed Comments To The Authors:**

The paper is pleasant to ready, although technical derivations are not always easy to follow. Anyway, they seem correct and the bound is clearly useful. I have some doubts about computational complexity of the method, and also the lack of the baselines (one suggestion is IMD, there maybe others). This could limit the practical applicability of the method for large n. Otherwise, a good work.

**Q7 Justification For Your Score:**

Good work, well written. It could be made better if they comments are addressed.

**Q9 Complying With Reviewing Instructions:**

1: Yes.

---

### Official Review · Reviewer_fa2k · 2022-04-13

**Q2(1) Originality/Novelty:** 3
**Q2(2) Significance/Impact:** 3
**Q2(3) Correctness/Technical Quality:** 3
**Q2(6) Clarity Of Writing:** 4
**Q6 Overall Score:** 7
**Q8 Confidence In Your Score:** 3

**Q1 Summary And Contributions:**

The paper discusses the orthogonal gromov wasserstein distance, which differs from original GW by optimizing over orthogonal matrices. Upper and lower bounds are proposed to compute OGW, and the lower bound is shown to be efficient. Extensions to graphs with different sizes and node features are introduced. Empirical results demonstrate the tightness of the proposed OGW bounds over the GW bounds, and OGW_{lb}'s effectiveness on graph and point cloud data.

**Q10 Ethical Concerns (Optional):**

No.

**Q2 Assessment Of The Paper:**

More detailed information regarding each of these aspects is given below:

**Q2(4) Quality Of Experiments (Optional):**

3: Good: The experimental evaluation is adequate, and the results convincingly support the main claims.

**Q2(5) Reproducibility:**

3: Good: Key resources (e.g., proofs, code, data) are available and key details (e.g., proofs, experimental setup) are sufficiently well-described for competent researchers to confidently reproduce the main results.

**Q3 Main Strengths:**

1. The paper introduces a lower bound to OGW, which is a practical alternative to GW and efficient to compute.
2. The paper is written clearly and should be understandable to general audiences with little background on the specific topic.

**Q4 Main Weakness:**

1. Despite the extensive application of GW (such as domain adaptation), the experiments in this paper is limited to somewhat "simple" cases (graph classification, which can also be done by graph neural networks nowadays), and barycenter problems. It would be nice to extend this to other cases where GW is successfully applied.

2. Since there are a lot of technical details involved, it may be slightly harder to pinpoint the key contribution of the paper (the upper and lower bounds of OGW in Section 2.2 among the part that was done by previous works (e.g. the bounds for GW). This is possibly due to a lack of a dedicated "related work" section. I don't think it is strictly necessary, but it could potentially better place the work in the literature.

**Q5 Detailed Comments To The Authors:**

Adding on point 1:
- This is a minor point, but it would be nice to see a runtime comparison between GW, OGW_lb and OGW_ub empirically, which gives more evidence for the O(n^3) versus O(n^5) argument.
- I don't think I see any experiments on OFGW despite a page dedicated to it; for completeness, I believe it is best to have an experiment on it (even a very simple one would suffice).

General comment:
- It is unclear how the conclusions in Table 2 is drawn (how are the results evaluated). For 6, all the methods should have "flipped" the barycenter; for 9, OGW_lb seems to be okay but OGW_ub should be "flipped".
- The OGW_lb is surprisingly tight in the experiments. I wonder if this means that the Q1 and Q2 have very similar optimal values (in Eq 9), and if so, is there an explanation to why this is the case?

**Q7 Justification For Your Score:**

I think the paper is technically solid. The motivation for using OGW as an alternative to GW, as well as the developed lower bounds are interesting and could be quite useful given its complexity advantage. That being said, the experiments are relatively limited (except for the bound tightness), and I somewhat feel that the potential of OGW_lb could have better demonstrated if it is applied to more practical problems.

**Q9 Complying With Reviewing Instructions:**

1: Yes.

---

### Official Review · Reviewer_KeEB · 2022-04-21

**Q2(1) Originality/Novelty:** 2
**Q2(2) Significance/Impact:** 3
**Q2(3) Correctness/Technical Quality:** 3
**Q2(6) Clarity Of Writing:** 4
**Q6 Overall Score:** 4
**Q8 Confidence In Your Score:** 4

**Q1 Summary And Contributions:**

The paper reviews relaxations of the optimization problem defining the Gromov-Wassertein (GW) distance on graphs, to obtain divergences that are tractable and can therefore be used in practice. It considers extensions to graphs with different number of nodes, and to include node features. It shows as well how the proposed divergences can be used to compute barycenters. Experiments illustrates the tightness of some bounds, applications to barycenter computations, and classification of graphs.


**Q2 Assessment Of The Paper:**

More detailed information regarding each of these aspects is given below:

**Q2(4) Quality Of Experiments (Optional):**

2: Fair: The experimental evaluation is weak: important baselines are missing, or the results do not adequately support the main claims.

**Q2(5) Reproducibility:**

2: Fair: Key resources (e.g., proofs, code, data) are unavailable but key details (e.g., proof sketches, experimental setup) are sufficiently well-described for an expert to confidently reproduce the main results.

**Q3 Main Strengths:**

- The paper make an interesting systematic investigations of the different relaxations available for GW and contribute a new one using the ideas of Hadley et al. (1992)
- The paper exploits these ideas rather thoroughly and applies to different tasks (barycenters, representation).
- The paper is well written
- The experiments are varied and interesting

**Q4 Main Weakness:**

- The experiments do not show empirically that the new relaxation proposed are significantly tighter than the ones appearing in the prior literature
- It is not obvious that the contributed relaxation (9) is tighter than the known relaxation (5) and this is not shown in the experiments.
- The paper is not showing that the relaxed divergences are distances or at least satisfy the triangle inequality, while this is needed for the kernel considered in the classification experiment to be really a kernel
- The proposed approach is in the end related to the work done in computer vision to propose kernels on 3D shapes in particular via relaxation (5) but this literature is not taken into account in the experiments.


**Q5 Detailed Comments To The Authors:**

1. The paper does not discuss when the relaxation proposed still yield distances (or at least a pseudo-distance) in the case of the $\ell_2$ losses considered. In particular, this is important because, if the triangle inequality is not satisfied, then the "kernel" considered in the experiments $\exp(-\gamma d(G,H))$ is not necessarily a kernel, in the sense that it is not necessarily a positive definite function, and so it cannot be used naively inside of an SVM for example.
For all the relaxations that remain inside of the orthogonal group (which the paper precisely considers), I believe that the proof of Memoli (2011) to show the triangle inequality goes through, and that the triangle inequality still holds.
But when the authors actually optimize over two rotation matrices as in (9) it is really not clear that what is produce is a distance or can be used to obtain a kernel.

2. How does the proposed approach handle disconnected graphs?

3. If, in equation (21), $C$ and $D$ are chosen to be distances as described just after equation (1), then padding the matrix with zeros is equivalent to saying that the additional nodes are at distance zero from each other and from all other nodes (which violates the triangle inequality). Shouldn't the padding remain compatible with a distance interpretation? How about doing the exact opposite, i.e. consider that there is a distance $d_{\max}$ and pad the matrix with that $d_{\max}$ distance?
Indeed, if we would like to compare a graph with $n$ nodes with a graph with $m<n$ nodes, the simplest way to augment the graph with $m$ nodes to make it of size $n$ is to add nodes that are are not connected to the rest of the graph, but if they are not connected then it makes sense to consider that the distances between themselves and between them and the nodes in the rest of the graph are large.

4. In the experiments, it is not clear what is/are the distance(s) which is/are considered in each experiment and which define the matrices $C$ and $D$. I would assume that the conclusion could vary substantially depending on the which distance is chosen (diffusion distance, geodesic distance,etc). The paper focuses a lot on the influence of this or that upper/lower bound, but in practice I would assume that the choice of the distance or similarity which produces the matrices $C$ and $D$ will be more crucial for any application.

5. I find it unfortunate that in the the experiments there is no direct comparison of the bounds from Memoli et al. ($GW_{xlb}$ and $GW_{xlb}$) with the orthogonal $(OGW_{zb})$ bounds. In particular $GW_{flb}$ seems to already be quite tight and its cost is only $O(n^2).$ So I would like to see in practice whether the bounds proposed are significantly better. Also, even though it is not possible to say in general whether $OGW_{ub}$ will be larger or smaller than $GW,$ it would be interesting to see how it behaves in practice.
The experiment presented on Figure 3.(c) concentrates on the gap between $OGW_{ub}$ and $OGW_{lb}$, but if $OGW$ is not a good approximation to $GW,$ what is the motivation for using $OGW$?

6. Also, it would be nice to provide an empirical comparison between the upper bound of equation (5) and the upper bound from equation (9). Indeed, going from maximizing over $\mathcal{O}$ to maximizing over $\mathcal{O} \cap \mathcal{E}$ there is a tightening of the bound. But given that there is no simple closed form expression for (8), the authors have to relax the bound to (9). So I would assume that there are actually cases where (9) is actually loser than (5).
I would suggest to plot as a scatter plot the gaps of (5) vs the gaps of (9) for the experiment of Figure 3 (c).

7. For the experiments on tightness, e.g. Figure 3, what does "randomly perturb" edges mean? Does it mean that you remove them? Change the distance? A more precise description would be needed for reproducibility.

8. For the classification problem considered, the paper makes comparison with a number of graph kernel which is certainly good. However, there is no comparison with the approaches that are conceptually the closest and that have been proposed in the computer vision literature in the context of shape analysis, in particular the heat kernel signature and the wave kernel signature.

   - J. Sun, M. Ovsjanikov, and L. Guibas, “A Concise and Provably Informative Multi-Scale Signature Based on Heat Diffusion,” in Computer Graphics Forum, vol. 28, no. 5, 2009, pp. 1383–1392.
   - M. Aubry, U. Schlickewei, and D. Cremers, “The wave kernel signature: a quantum mechanical approach to shape analyis,” in Proc. CVPR, 2011.

   Another even simpler representation which would be very natural in the context of this work is a representation which is the simplest representation that is invariant w.r.t. permutations (and also to rotations) and which is to represent the matrices C and D as the sorted vector of their eigenvalues.  This leads for example to consider the Gaussian kernel on the spectra of C and D, which is obviously directly related to relaxation (5).

9. "Rendl and Wolkowicz (1992) first pointed out that [...] can be maximized exactly by a simple eigendecomposition"
$\rightarrow$ it was essentially already described in

   - Umeyama, S. (1988). An eigendecomposition approach to weighted graph matching problems. IEEE transactions on pattern analysis and machine intelligence, 10(5), 695-703.

10. The text says "$c_G(i,j)$ represents a distance measure on $G$" but in which sense does it have to be a distance "on G"? My understanding is that it is assumed to be a distance in some space but not "on G".  (Of course, as the authors mention it later it can be other things than a distance including a kernel matrix...)

11. "Noting that neither triangle inequality nor symmetry in $G$ and $H$ is so relevant in many applications, we will follow their framework and simply refer to it as GW" $\rightarrow$
I am surprised by this sentence: given that the rest of the paper is actually focussing on the $\ell_2$ case, and that in that case, as pointed out by Peyre et al.,
    - GW always satisfies the triangle inequality (so it is always a pseudo-metric) and
    - it is equal to zero if and only if there is an isometry, as soon as the matrices C and D corresponding to $c_{G}$ and $c_{H}$ are positive semi-definite or distance matrices.

    So, it seems to me that throughout this paper it is only the case of the GW distance (and not divergence) that is considered and that discussing the possibility of using another loss than L2 is confusing here, because it does not match with the focus of the rest of the paper.

12. Minor comments on the presentation:
   - The color legend in Figure 3 is really too small. Even with the paper displayed at A2 scale on my monitor it was still small.
   -  It would help the reader if captions of Figure 3 were more complete, given that subplots are associated with different experiments.


**Q7 Justification For Your Score:**

- The authors clearly put some efforts in the experiments, but one serious issue is that the experiments are not showing that the new bounds proposed in the paper are really useful in the sense that they woud be really tighter than what has been proposed before. The paper shows that the gap between the upper and lower bounds on OGW are tighter, if the gap between OGW and GW is large, then does it make really sense to be using OGW_{ub}.
- (9) does not yield a distance, while (5) does...

**Q9 Complying With Reviewing Instructions:**

1: Yes.

---

### Decision · Program_Chairs · 2022-05-15

**Decision:**

Accept (Poster)

**Comment:**

Meta Review: The submission is centered around similarity of graphs, specifically the Gromov-Wasserstein (GW) distance. Motivated by the computational difficulty of GW, the authors propose the orthogonal GW [(6)], which gives rise to computationally tractable lower bound, with extension to graphs having different number of nodes and to ones where the nodes are equipped with features. The idea is illustrated in the context of graph classification and barycenter problems.

Studying similarity measures of graphs is a central problem of machine learning. The authors propose principled tools in this context  which can be of both theoretical and practical interest to the machine learning community, as it was assessed by the reviewers.